# Rapid Detection of Avocado Oil Adulteration Using Low-Field Nuclear Magnetic Resonance

**DOI:** 10.3390/foods11081134

**Published:** 2022-04-14

**Authors:** Haoquan Jin, Yuxuan Wang, Bowen Lv, Kexin Zhang, Zhe Zhu, Di Zhao, Chunbao Li

**Affiliations:** Key Laboratory of Meat Processing, MOA, Key Laboratory of Meat Processing and Quality Control, MOE, Jiang Synergetic Innovation Center of Meat Production, Processing and Quality Control, Nanjing Agricultural University, Nanjing 210095, China; 2019108027@njau.edu.cn (H.J.); plezam@outlook.com (Y.W.); 9201810407@stu.njau.edu.cn (B.L.); 9201810417@stu.njau.edu.cn (K.Z.); 9201810408@stu.njau.edu.cn (Z.Z.); chunbao.li@njau.edu.cn (C.L.)

**Keywords:** low field nuclear magnetic resonance, avocado oil, adulteration detection, chemometrics

## Abstract

Avocado oil (AO) has been found to be adulterated by low-price oil in the market, calling for an efficient method to detect the authenticity of AO. In this work, a rapid and nondestructive method was developed to detect adulterated AO based on low-field nuclear magnetic resonance (LF-NMR, 43 MHz) detection and chemometrics analysis. PCA analysis revealed that the relaxation components area (S_2__3_) and relative contribution (P_22_ and P_2__3_) were crucial LF-NMR parameters to distinguish AO from AO adulterated by soybean oil (SO), corn oil (CO) or rapeseed oil (RO). A Soft Independent Modelling of Class Analogy (SIMCA) model was established to identify the types of adulterated oils with a high calibration (0.98) and validation accuracy (0.93). Compared with partial least squares regression (PLSR) models, the support vector regression (SVR) model showed better prediction performance to calculate the adulteration levels when AO was adulterated by SO, CO and RO, with high square correlation coefficient of calibration (R^2^_C_ > 0.98) and low root mean square error of calibration (RMSEC < 0.04) as well as root mean square error of prediction (RMSEP < 0.09) values. Compared with SO- and CO-adulterated AO, RO-adulterated AO was more difficult to detect due to the greatest similarity in fatty acids’ composition being between AO and RO, which is characterized by the high level of monounsaturated fatty acids and viscosity. This study could provide an effective method for detecting the authenticity of AO.

## 1. Introduction

Avocados, which contain a high level of oil, are widely grown in New Zealand, Mexico, Chile, South Africa and parts of China. Avocado oil (AO) is widely applied in the pharmaceutical and cosmetic industries [1] due to its high level of bioactive compounds in AO, including tocopherols and carotenoids [2,3,4]. In addition, AO is rich in monounsaturated fatty acids (MUFA), such as oleic acid, which are beneficial to the cardiovascular system and help to enhance the absorption of nonpolar functional compounds and to prevent cancers [5,6,7]. AO is also consumed as an edible oil due to its high nutrition values [8]. Compared with commonly-consumed corn oil (CO), soybean oil (SO) and rapeseed oil (RO), AO is a relatively higher price and has a risk of being adulterated [9,10]. In addition, the geographical origin and the botanical variety are also an important consideration in the authenticity of commercial oil [11,12].

Various methods have been developed to identify the adulteration of edible oil by detecting the fatty acid patterns of tested samples, including gas chromatography (GC) [13], gas chromatography-mass spectrometry (GC-MS) [14,15,16] and high-performance liquid chromatography (HPLC) protocols [17,18]. Despite the high sensitivity, these methods usually either consume a lot of time or involve the application of toxic reagents, rendering them unsuitable for rapid detection. Recently, a near infrared spectroscopy (NIR) method, a rapid and nondestructive method, was established to identify and quantify avocado oil adulteration [19,20].

Low field nuclear magnetic resonance (LF-NMR) is a nondestructive, rapid and environmentally friendly detection method, whereas LF-NMR has a disadvantage in its detection limit. Different from high field NMR that can detect a compound based on the chemical shift of H or C in this compound, LF-NMR can usually detect a compound according to the attenuation characteristics of the H of the compound, which is called the relaxometry signal. The relaxometry signal of a compound is not only determined by its structure but also by the microenvironment surrounding the compound [21]. Currently, LF-NMR is applied in the analysis of food properties, especially in detecting the changes in water and oil fractions [22,23]. Bound water, immobile water and free water can be well distinguished in a relaxometry measurement by LF-NMR, which is a typical application [24]. Compact low field NMR spectroscopy and chemometrics can be applied to the analysis of edible oils [25]. Considering that oil adulteration usually occurs at high adulteration levels, LF-NMR could be a suitable method for detecting AO adulteration by cheaper oils. Therefore, this work evaluated whether the LF-NMR parameters of AO and commonly-consumed oils (CO, SO and RO) have sufficient characteristics to be distinguished in a principal component analysis (PCA) model. In addition, the LF-NMR parameters of all pure AO and adulterated AO were collected, and subsequently input to the Soft Independent Modelling of Class Analogy (SIMCA), partial least squares regression (PLSR) and support vector regression (SVR) models to develop either qualitative or quantitative analysis models. In addition, the fatty acids’ composition and viscosity of tested oils were investigated to reveal the underlying basis for the different LF-NMR parameters of AO from SO, CO and RO. This work may provide a new method for the rapid identification of AO adulteration.

## 2. Materials and Methods

### 2.1. Preparation of Oil Samples

Pure sample extra virgin cold pressed AO (4 brands), which were produced in France, New Zealand or Mexico, were purchased in local Suguo supermarket (Nanjing, China); each brand included five independent bottles of oil. SO, CO and RO samples were obtained from oil manufacturer A (Qinhuangdao, China). The adulterated oil samples were prepared by adding SO, CO or RO into AO at percentages of 0, 10, 20, 30, 40, 60, 80, 100% (*w*/*w*) to prepare SO-, CO- and RO-adulterated AO samples (AO-SO, AO-CO and AO-RO); each sample was prepared in 10 replicates. The AO used to prepare the AO-SO, AO-CO and AO-RO samples was randomly selected from four brands. As shown in Appendix A, 4 brands of AO (n = 20), 1 brand of SO (n = 5), 1 brand of CO (n = 5), 1 brand of RO (n = 5), 6 concentration levels (10–80%) of AO-SO samples (n = 10), 6 concentration levels (10–80%) of AO-CO samples (n = 10) and 6 concentration levels (10–80%) of AO-RO samples (n = 10) were used for the following LF-NMR detection. Each sample was loaded in a 7-mL screw-capped glass vial and preserved in a refrigerator at 4 °C before LF-NMR detection.

### 2.2. Acquisition of LF-NMR Signals

The ^1^H relaxation time curve of the oils was acquired on a 43 MHz NMI20-040H-I LF-NMR spectrometer (Niumag Corporation, Suzhou, China). During measurement, each sample (5 g) was transferred into the probe (40 mm). The Carr–Purcell–Meiboom–Gill sequence (sampling frequency = 200 KHz, repeated waiting time = 5000 ms, echo count = 8000, the time between 90° and 180° pulse was 0.1 ms and repeat scan times = 8) was applied to detect the transverse relaxation time (T_2_) of each sample. All the oil samples were transferred to a thermostatic water bath and equilibrated to 32 °C, and then were placed in the probe for 1 min before sampling [26]. As shown in Figure 1, the relaxation time of three characteristic relaxation components was determined as T_21_, T_22_ and T_23_. The start time of relaxation components (T_21S_, T_22S_ and T_23S_), peak time of relaxation components (T_21P_, T_22P_ and T_23P_), end time of relaxation components (T_21E_, T_22E_ and T_23E_), single component relaxation time (T_2W_, inverted from CPMG decay data using a single-exponential fitting), percentage relative contribution (P_21_, P_22_ and P_23_), relaxation components area (S_21_, S_22_ and S_23_) of each relaxation components and total area (S_Total_) of all relaxation components were collected as key parameters for the following chemometrics analysis.

### 2.3. Viscosity Measurement

Oil viscosity of each sample was measured using a Physical MCR301 rheometer (Anton Paar, Graz, Austria) with a parallel-plate (radius = 25 mm) geometry. The gap between the plates was 1.0 mm and the measurement was applied at 25 °C. Data were obtained using the shear rate from 0.1 to 1000 s^−1^. Each measurement was performed in quintuplicate to obtain an average value.

### 2.4. Analysis of Fatty Acid Composition

Fatty acid methyl esters mixed standard (37 types) was purchased from Yuanye Biochemical Co., Ltd. (Shanghai, China) to qualify the fatty acid composition of tested oil samples. The methyl esterification of the fatty acids of commercial oil referred to the method of Amit et al. with little modification [27]. Briefly, 0.1 g of oil sample was boiled with 4 mL 2% methanolic sodium hydroxide at 70 °C for 10 min in glass vials. The vials were added in 5 mL boron trifluoride 14% methanol and were again boiled for 5 min. The vials were cooled on ice, and then 10 mL N-heptane solution and 5 mL of saturated sodium chloride were added. The vials were shaken vigorously and left to stand for the layer separation. The upper layer containing methyl esters was transferred to GC vials. Fatty acids were determined using a GC-2010 plus gas chromatography (Shimadzu, Kyoto, Japan) equipped with a flame ionization detector and a 2560 capillary column (100 m × 0.25 mm, 0.2 µm; Supelco, Sigma-Aldrich, Bellefonte, PA, USA). The operation parameters were as follows: 1 mL/min of nitrogen was supplied; samples (1.0 µL) were loaded in a split ratio of 1:100 at 270 °C of inlet temperature and at 280 °C of detector temperature. The oven temperature was programmed as follows: 100 °C (15 min), 180 °C (10 °C/min, 6 min), 200 °C (1 °C/min, 20 min), 230 °C (4 °C/min, 10.5 min). Relative content (%) of fatty acids was calculated as the ratio between the peak area of selected fatty acid and the total area of all peaks. Each measurement was performed in quintuplicate to obtain an average value.

### 2.5. Statistics Analysis

Significance analysis: Statistical differences were analyzed by one-way analysis of variance under the Duncan’s multiple range tests in SAS 8.0 (SAS Institute Inc., Cary, NC, USA). Differences were significant when the *p*-value was smaller than 0.05. Figures were made in Origin Pro 2016 (Origin Lab, Northampton, MA, USA).

Chemometrics analysis: Before the development of these models, all the samples were separated into calibration sets and validation sets randomly. Each calibration set (141 samples) was used for mathematical modelling, and another 59 samples were applied as the validation set to evaluate the predictive ability of the following models. The original dataset, including LF-NMR parameters T_2W_, T_21S_, T_21P_, T_21E_, T_22S_, T_22P_, T_22E_, T_23S_, T_23P_, T_23E_, S_21_, S_22_, S_23_, S_Total_, P_21_, P_22_ and P_23_ were input into SIMCA version 14.1 (Umetrics AB, Umea, Sweden) for PCA, SIMCA, PLSR and SVR analyses. The SVR analysis was processed in MATLAB R2016b (The Math Works Inc., Natick, MA, USA).

The LF-NMR parameters from pure AO, AO-SO, AO-CO and AO-RO in Appendix A were used in PCA and SIMCA analysis. PCA, an unsupervised multivariate method, was applied to initially explain the most variability in the original datasets [22]. SIMCA was used to establish models for distinguishing each population (AO, AO-SO, AO-CO and AO-RO). The distance between a point and the centroid (Mahalanobis distance, DModXPS+) was applied to identify the outliers of each population. When the DModXPS+ for a certain sample was larger than the critical limit (DCrit, 95% confidence interval), the sample was considered as an outlier. The prediction performance of the SIMCA model was evaluated via true positive rate (*TPR*), false negative rate (*FNR*), positive predictive value (*PPV*), false discovery rate (*FDR*) and accuracy
$$TPR=\frac{TP}{TP+FN}$$
$$FNR=\frac{FN}{FN+TN}$$
$$PPV=\frac{TP}{TP+FP}$$
$$FDR=\frac{FP}{TP+FP}$$
$$Accuracy=\frac{TP+TN}{TP+FN+FP+TN}$$
where *FN* is the number of false negatives, *F**P* is the number of false positives, *TN* is the number of true negatives and *TP* is the number of true positives.

The LF-NMR parameters from all the samples in Appendix A were used in PLSR and SVR analysis. The supervised PLSR model was founded by a calibration set with a small number of latent variables. The number of latent variables selected for PLSR model was chosen according to the root mean square error of cross validation (RMSECV), carried out by 7-fold cross validation. The PLSR and SVR models were evaluated by considering the square correlation coefficient of calibration and validation between the predicted and observed value (R^2^_C_ and R^2^_P_), root mean square error of calibration (RMSEC), RMSECV and root mean square error of prediction (RMSEP) [28].

## 3. Results and Discussion

### 3.1. Comparison of T_2_ Parameters between Pure and Adulterated AO

Figure 1 shows the T_2_ distribution of AO adulterated with different levels (0–100%) of SO, CO or RO. Three typical relaxation components labeled as T_21_ (2.13–3.40 ms), T_22_ (65.03–72.71 ms) and T_23_ (209.59–251.98 ms) were identified in tested oil samples. Compared with pure AO, the T_23_ component of AO-SO relaxes more slowly but its T_21_ component relaxes faster (Figure 1A,B), whereas RO-adulterated AO was shown to share a similar relaxation time with AO (Figure 1C). These results were due to the similarity in the fatty acid compositions between RO and AO, which was confirmed in the following analysis of the fatty acids’ composition. As shown in Appendix A and Appendix A, the values of T_2__W_ (116.6–147.9 for SO and 119.3–144.1 for CO), T_22P_ (65.0–72.7 for SO and 65.9–71.3 for CO) and T_23P_ (209.6–252.0 for SO and 214.8–243.7 for CO) gradually increased along with the increase (10–100%) in the adulteration level of CO or SO. The P_2__2_ values gradually decreased along with the increase in the adulteration levels (10–100%) of SO (61.1–55.5) or CO (61.1–55.1), whereas the P_23_ values gradually increased along the with the increase in the adulteration levels of SO (36.6–42.7) or CO (36.8–43.0). In addition, adulterated oils were generally found to possess lower P_22_ but higher P_23_ values compared with the pure AO. However, few significant differences were found in LF-NMR parameters when adulteration levels of SO and CO were smaller than 30% and 40%, respectively. Based on these basic parameters, AO adulterated by SO or CO was easier to identify than that adulterated by RO. At a lower adulteration level, adulterated AO and pure AO were difficult to be distinguished. Therefore, suitable multivariate analysis methods, including PCA, SIMCA, PLSR and SVR, were introduced to distinguish or predict adulteration levels and are described in the following sections.

### 3.2. Classification of Pure and Adulterated AO by PCA and SIMCA Analyses

PCA was used as an unsupervised multivariate tool to estimate whether pure AO can be distinguished from SO-, CO- and RO-adulterated AO. LF-NMR parameters of pure and adulterated AO, including T_2W_, T_21S_, T_21P_, T_21E_, T_22S_, T_22P_, T_22E_, T_23S_, T_23P_, T_23E_, S_21_, S_22_, S_23_, S_Total_, P_21_, P_22_ and P_23_ were collected and normalized using a unit variance scaling (UV) method in SIMCA version 14.1 and subjected to the PCA model. The data used to build the PCA models were shown in Appendix A. The score plot and loading plot of PCA are shown in Figure 2. The PCA model score plot (Figure 2A) showed that the pure AO can be largely separated from SO- and CO-adulterated AO, whereas AO dots overlapped with the dots of some RO-adulterated samples. In the loading plot (Figure 2B), P_23_, T_2W_, S_2__2_, P_22_ and S_23_ are shown to be key original variables in the principal component 1 (PC1), and T_21S_, T_21P_, T_21E_, T_22E_ and T_23S_ are key original variables in the PC2 to discriminate AO from adulterated AO. In the PCA model, it was still difficult to determine where tested samples belonged when the points in the PCA plots were close to each other. Therefore, the SIMCA model was established, as described in the following section, to distinguish AO from other samples.

SIMCA is a discrimination analysis tool that builds an independent PCA model for each category of samples (Appendix A) and determines the attribution according to the distance from the calibration set to the model [29,30]. The first step was to use the Mahalanobis distance to detect outliers in each of the four groups (Appendix A). No outlier was detected in any of the four groups (AO, AO-SO, AO-CO and AO-RO). . The all data used to build the SIMCA models were shown in Appendix A. Fifteen pure AO samples, 42 AO-SO samples (adulteration levels from 10 to 80%), 42 AO-CO samples (adulteration levels from 10 to 80%) and 42 AO-RO samples (adulteration levels from 10% to 80%) were randomly selected to establish SIMCA models for calibration in Figure 3A. Five pure AO samples, 18 AO-SO samples, 18 AO-CO samples and 18 AO-RO samples were selected for validation in Figure 3B. In the SIMCA for pure AO (Figure 3A,B), the line of DCrit_0.05_ = 2.091 is the discrimination threshold of the pure AO samples. The samples with DModXPS+ values under DCrit_0.05_ were identified as AO. Samples with values slightly higher than the critical line but closer to the AO cluster in the PCA model were also identified as AO. The remaining samples were identified as adulterated AO. In Figure 3C–H, the SIMCA models of AO-SO, AO-CO and AO-RO are analyzed using a similar principle to that of AO. The DCrit_0.05_ values for the AO-SO, AO-CO and AO-RO models were 1.548, 1.522 and 1.522, respectively. The performance of the SIMCA models was tested by establishing receiver operating characteristic (ROC) curves on the basis of predictions of the cross validation procedure (Appendix A). The accuracy of each SIMCA model was judged by the area under the curve (AUC), which was greater than 0.963 in all models (Table 1). The established SIMCA models showed good performance in the following parameters: the measurement of fit (R^2^_X_) > 0.993 and measurement of prediction (Q^2^) > 0.886. The closer these values are to 1, the better the model. In addition, Figure 3F,I show the high accuracy of the models in both calibration (97.87%) and validation sets (93.22%). Table 2 shows an excellent prediction performance based on the calculated TPR (>0.93), FNR (<0.07), PPV (>0.95) and FDR (<0.05) in the calibration sets, and TPR (>0.6), FNR (<0.4), PPV (>0.89) and FDR (<0.11) in the validation sets. In the SIMCA model, SO-, CO- and RO-adulterated AO is easily detected when the adulteration level reaches 20%. The SIMCA model cannot only effectively identify whether AO was adulterated, but can also determine which type of oil was added to AO.

### 3.3. Quantitative Analysis of Adulteration Levels by PLSR and SVR

PLSR was used to build a regression model to predict the adulteration level of CO, SO and RO in adulterated AO. A PLSR model is comprised of calibration and validation parts. In the construction of the calibration model, LF-NMR parameters of the calibration set were used to obtain a regression equation between the parameters and the adulteration levels. The data used to build the PLSR models were shown in Appendix A. In the validation section, another external dataset was imported into the equations to calculate the adulteration level of the samples [31]. The relationship between the number of latent variables and RMSECV was found using a 7-fold cross validation approach to determine the optimum number of latent variables [32]. This procedure is a crucial way to decrease the RMSECV value, since an elevation in the number of latent variables may result in an over-fitting and low predictive power but fewer latent variables could result in under-fitting and information loss [33]. The comparative significance of the founded model was evaluated by calculating RMSECV, RMSEC and R^2^ values. The model with lower RMSEC and RMSECV values and a higher R^2^_C_ value is usually considered to be better. The predictive ability of the established model was evaluated by calculating RMSEP and R^2^_P_ values. The model with lower RMSEP and higher R^2^_P_ values is a relatively better one [19,32].

Based on the LF-NMR characteristic parameters of SO-, CO- and RO-adulterated AO samples, the PLSR quantitative analysis models were established, and the results are shown in Figure 4A,C,E. The key parameters of models, including RMSEC, RMSECV, R^2^_C_, RMSEP and R^2^_P_ are also shown in Figure 4A,C,E. Based on changes in RMSECV values, three (AO-SO model), three (AO-CO model) and two (AO-RO model) latent variables were selected for the three PLSR models since RMSECV did not decrease significantly along with an increase in latent variable. The performance of the calibration model can be evaluated by the RMSEC and R^2^_C_ values, whereas RMSEP and R^2^_P_ can be used to evaluate the performance of the validation model. PLSR analysis of AO-SO and AO-CO showed better prediction performance than that of AO-RO in both the calibration and validation models. The R^2^_C_ and R^2^_P_ values of the AO-SO and AO-CO models exceeded 0.962 and 0.969, and the values of RMSEC, RMSECV and RMSEP were less than 0.061, 0.071, and 0.57, respectively. In contrast, the quantitative analysis model of AO-RO had lower R2C (0.827) and R2P (0.666) values, and higher RMSEC (0.130), RMSECV (0.145) and RMSEP values (0.173). According to the results of the standardized coefficients (Appendix A), T_2W_, T_22S_, T_22P_, T_23P_, T_23E_, P_22_ and P_23_ were key in PLSR models of AO-SO and AO-CO, and S_21_, T_22S_, T_22E_, T_23S_, T_23E_ and P_21_ played an important role in the PLSR model of AO-RO.

In order to establish a reliable model to predict the adulteration level of RO in adulterated AO, we checked the model’s linearity using the Durbin–Watson (*DW*) test. This study estimated the probability associated with the *DW* statistics (Equation (1)), where *p* > 0.05 indicates the linear and *p* < 0.05 nonlinear behavior [34].
(1)$$DW=\frac{\sum_{i=2}^{n}{\left(e_{i}-e_{i-1}\right)}^{2}}{\sum_{i=1}^{n}e_{i}^{2}}$$
where $e_{i}$ is the ith residual and n is the number of calibration samples. The *p*-value of the *DW* test was 0.0071 (<0.05), which indicated the PLSR model of AO-RO obeyed nonlinearity. Consequently, the nonlinear SVR analysis was applied and the result is shown in Figure 4F. Compared with the PLSR model, the SVR model showed much better prediction performance in the quantitative analysis of RO-adulterated AO. The SVR model of AO-RO offered the higher R^2^ (R^2^_C_ = 0.978, R^2^_P_ = 0.892) and the lower RMSE (RMSEC = 0.047, RMSECV = 0.078, RMSEP = 0.109). In addition, the SVR analysis of SO-adulterated AO (Figure 4B) and CO-adulterated AO (Figure 4D) also showed better performance than PLSR analysis since higher R^2^_C_ and R^2^_P_ values as well as lower RMSEC, RMSECV and RMSEP values were obtained in SVR models.

In real cases of oil authenticity issues, the oil used as an adulterant is not known. Therefore, the class is composed of pure AO and all the adulterated oil combined into one class. The LF-NMR parameters of SO-, CO- or RO-adulterated AO samples (AO-SO/CO/RO) were randomly input into software to establish PLSR and SVR models, in which the level of AO and adulterant oil can be calculated regardless of the kind of oil that was added to AO. Again, the SVR model showed better performance (R^2^_C_ = 0.986, R^2^_P_ = 0.907, RMSEC = 0.035, RMSECV = 0.064 and RMSEP = 0.089) than PLSR analysis. Based on these results, an LF-NMR database established using an SVR model could be a promising way to predict the alteration level of AO by oils with a lower price.

### 3.4. Comparison of Viscosity and Fatty Acid Composition

Both qualitative and quantitative models were established in previous sections, and the underlying mechanism was studied in this section. A comparison of the T_2_ distribution of five pure tested oils is shown in Figure 5A. The T_22_ relaxation component area (S_22_) of AO was the largest, whereas their T_23_ area and relaxation time were the smallest among all of the oils. Compared with AO, the LF-NMR curves of SO, CO and RO shifted to the right direction. To investigate where the differences in the LF-NMR curves between AO and the rest of the oils come from, the viscosity and fatty acid composition of these oils were determined.

The viscosity of the four oils is illustrated in Figure 5B. The descending order of viscosity was AO > RO > CO > SO. The fatty acid composition of the four pure oils is illustrated in Table 3. The major fatty acids in AO were oleic oil (C18:1n9c, >64.49%), linoleic oil (C18:2n6c, >11.77%) and palmitic (C16:0, >10.66%). The fatty acid profile of AO was shown to be of a MUFA > polyunsaturated fatty acids (PUFA) > saturated fatty acids (SFA) pattern, and RO shared a similar order to AO. In contrast, the SO and CO followed a PUFA > MUFA > SFA pattern. Therefore, with the increase in the adulteration levels of SO and CO into AO, the oleic acid content decreased and the linoleic acid content increased gradually.

In the PCA score plot (Figure 5C), AO points with lower unsaturation levels are distributed in the positive quadrant of the PC1, whereas SO and CO points with higher unsaturation levels are distributed in the negative quadrant. Moreover, the result in Figure 5D shows that the viscosity is positively correlated with the value of MUFA but is negatively correlated with the value of PUFA. The content of PUFAs in AO was the lowest among the four tested oils, which could account for the lower level of hydrogen protons in this sample. With the decrease in the degree of unsaturation of fatty acids, the binding force of hydrogen protons increases, resulting in a speeding up of the decay of the LF-NMR signal and shortening of the relaxation time [35]. The LF-NMR parameters of the four pure oils were analyzed, The PCA score plot and loading plot are shown in Figure 5E,F. The cumulative variance contribution rate reached 88.7%, indicating that the PCA model can reflect most of the information of total variations. S_23_, P_22_, P_23_, S_Total_ and T_2W_ were crucial parameters to distinguish AO from the adulterated AO, and T_21S_, T_21P_ and T_21E_ were helpful in distinguishing AO from RO. The parameters of the second relaxation component (S_22_ and P_22_) and the parameters of the third relaxation component (S_23_ and P_23_) of T_2_ spectra were related to the number of relatively stable and unstable hydrogen protons, respectively [36]. Therefore, the differences in LF-NMR parameters between AO, SO, CO and RO were mainly reflected in the values of S_22_, P_22_, S_Total_ and T_23P_, and the reason for the differences was the different fatty acid composition of oils. In a word, the higher the saturated level of oil, the larger the viscosity of oil and the shorter the relaxation time. In addition, a negative relationship between the polarity of oil and relaxometry time in the T_2_ area, and a positive relationship between the relaxometry intensity and hydrogen atom density have been reported [37,38,39]. Therefore, the discrepancies in the polarity and hydrogen atom density between AO, SO, CO and RO could also account for their differences in the relaxometry time and intensity, which contribute to the qualitative and quantitative analysis of AO in this work. In real practice, LF-NMR can be used as a rapid screening to find adulteration candidates, and then other time-consuming but more accurate methods such as GC and GC-MS can be applied to obtain the final conclusion.

## 4. Conclusions

This work explored the possibility of applying an LF-NMR measurement and chemometrics model in the detection of AO adulteration. In SIMCA models, LF-NMR allowed for the successful classification of AO and SO-, CO- and RO-adulterated oil with an accuracy of 98%. Furthermore, PLSR and SVR models offered reliable techniques for the quantification of SO, CO or RO that were adulterated in AO, as indicated by the high levels of R^2^ and the low level of RMSE in both calibration and validation models. Compared with SO- and CO-adulterated AO, RO-adulterated AO was more difficult to distinguish from AO due to their similar composition of fatty acids. Based on these results, LF-NMR could be a promising method in the rapid detection of oil adulteration, even though the models established remain to be verified in more brands of SO, CO, RO and other kinds of oil with a lower price.

## Figures and Tables

**Figure 1 foods-11-01134-f001:**
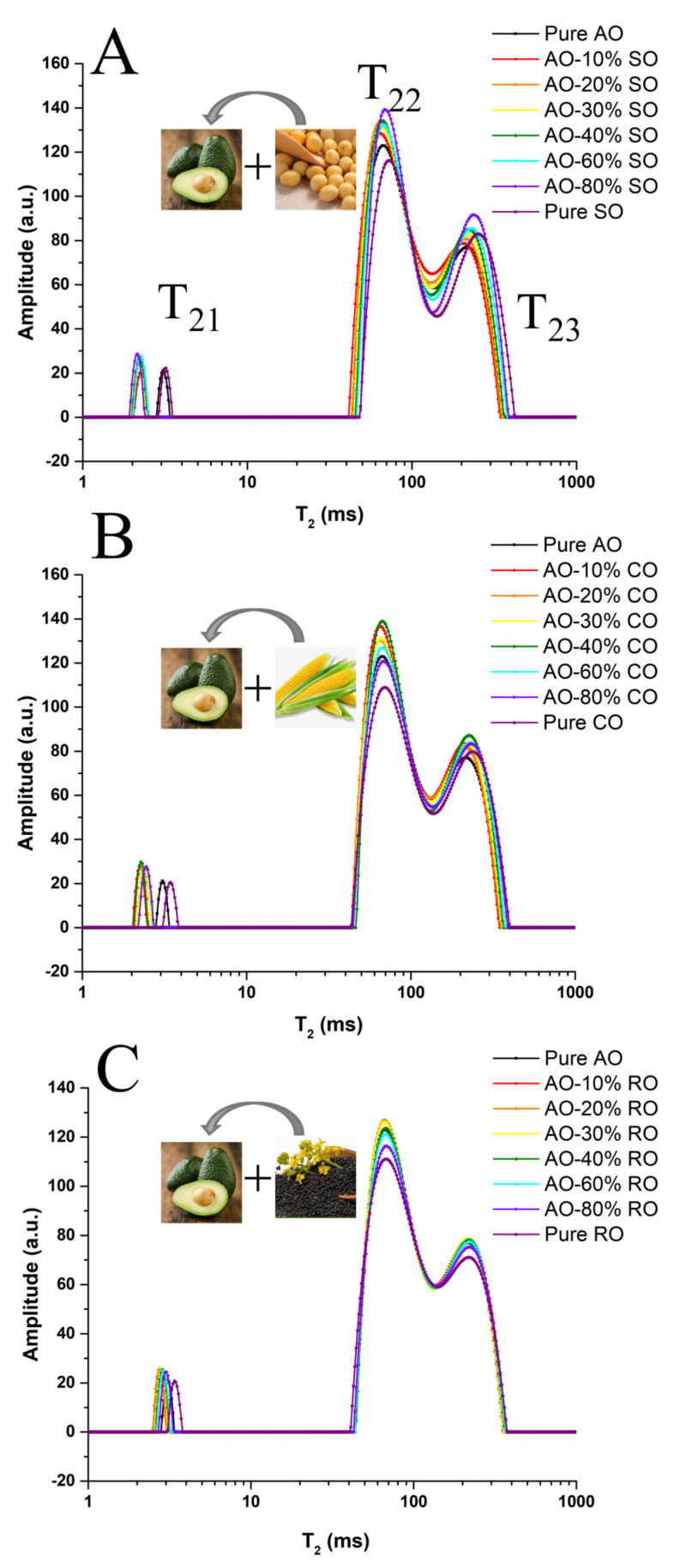
The T_2_ distribution of AO adulterated by SO (**A**), CO (**B**) or RO (**C**) (0–100%).

**Figure 2 foods-11-01134-f002:**
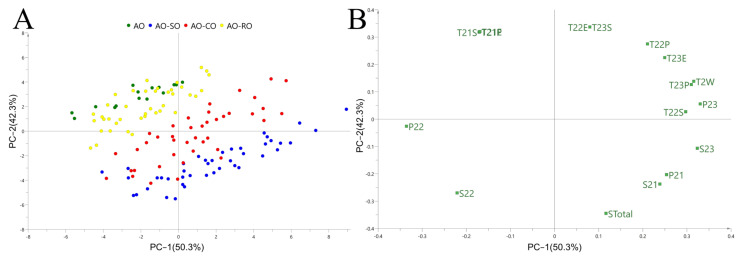
The PCA score plot (**A**) and loading plot (**B**) to distinguish pure AO from adulterated AO.

**Figure 3 foods-11-01134-f003:**
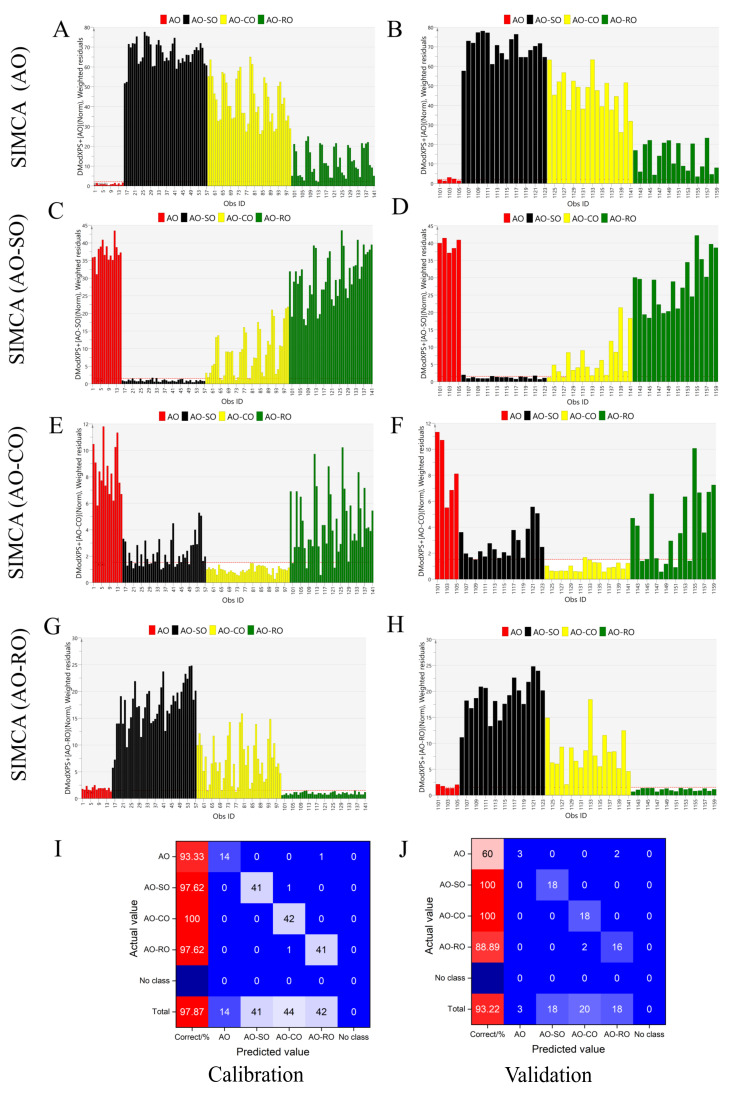
SIMCA analysis to distinguish pure AO from adulterated AO: The DModXPS+ diagram of pure AO (**A**,**B**), AO-SO (**C**,**D**), AO-CO (**E**,**F**) and AO-RO (**G**,**H**); the samples on the left (**A**,**C**,**E**,**G**) were used for calibration and the samples on the right (**B**,**D**,**F**,**H**) were used for validation; (**I**,**J**) show the specific discrimination results of calibration (**I**) and validation (**J**) in the form of confusion matrix.

**Figure 4 foods-11-01134-f004:**
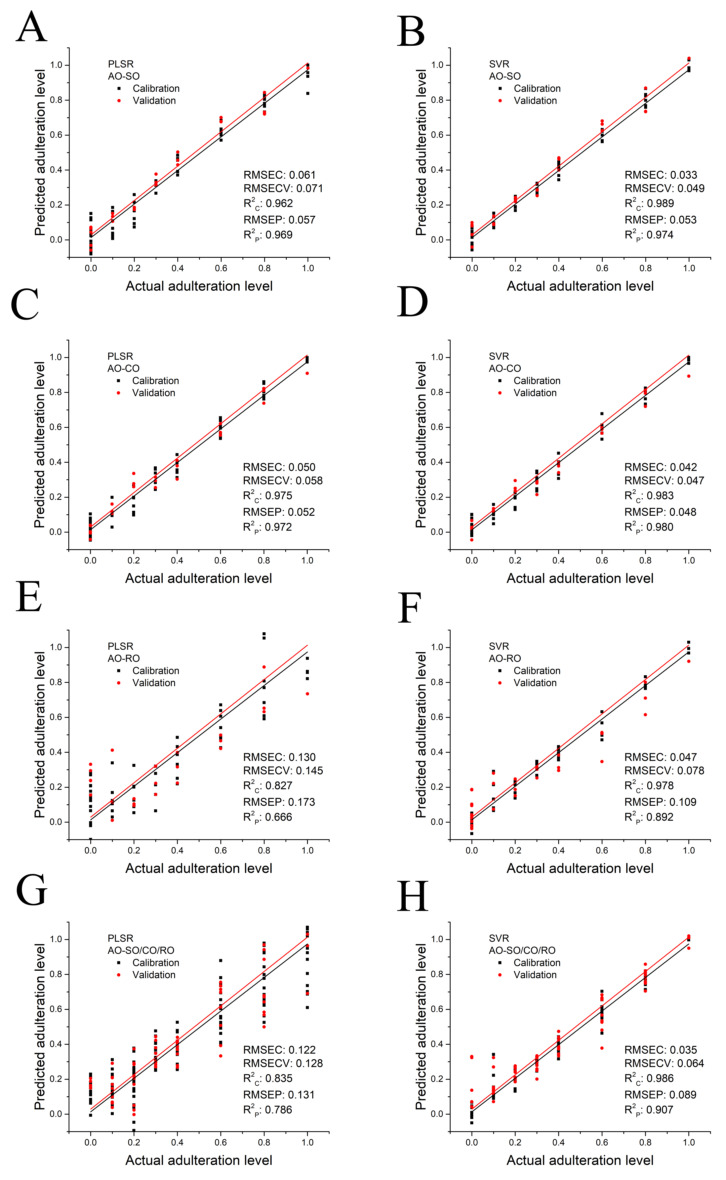
Prediction of adulteration level of AO adulterated by SO, CO or RO, as determined by the PLSR and SVR algorithm: AO-SO–AO adulterated with SO (**A**,**B**), AO-CO– AO adulterated with CO (**C**,**D**), AO-RO–AO adulterated with RO (**E**,**F**) and AO-SO/CO/RO–AO adulterated with SO, CO or RO randomly (**G**,**H**).

**Figure 5 foods-11-01134-f005:**
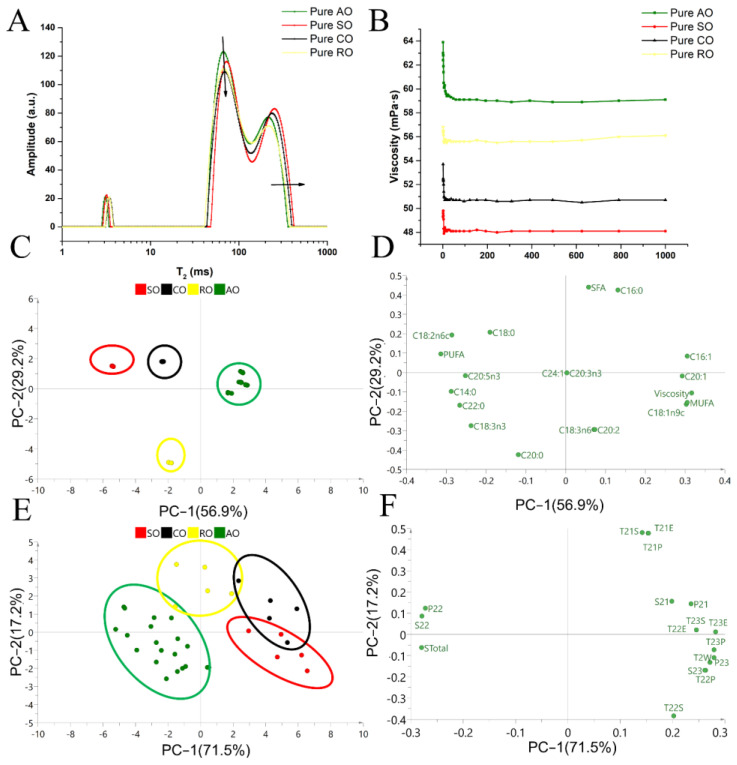
Analysis of fatty acids’ composition and viscosity of AO, SO, CO and RO to explain their differences in LF-NMR signals: (**A**) indicates T_2_ distribution of each pure oil; (**B**) indicates the viscosity of each pure oil; (**C**,**D**) indicate the PCA score plot and loading plot of pure oils based on data of fatty acid composition and viscosity; (**E**,**F**) indicate the PCA score plot and loading plot of four pure oils based on T_2_ parameters of LF-NMR profiles.

**Table 1 foods-11-01134-t001:** The model parameters of PCA and SIMCA.

Model	Number	PCs	R^2^_X_	Q^2^	AUC
PCA	141	6	0.995	0.981	-
SIMCA (AO)	15	7	0.996	0.886	1.000
SIMCA (AO-SO)	42	6	0.994	0.975	0.993
SIMCA (AO-CO)	42	5	0.993	0.974	0.963
SIMCA (AO-RO)	42	5	0.998	0.945	0.997

Note: PCs: principle components, R^2^_X_: the measurement of fit for the PCA or SIMCA model, Q^2^: the measurement of prediction, which is calculated through cross validation to evaluate the predictive ability of the model. AUC (area under curve) is the area enclosed by the coordinate axis under the ROC (receiver operating characteristic) curve.

**Table 2 foods-11-01134-t002:** Evaluation of the prediction performance of SIMCA models based on the parameters TPR, FNR, PPV, FDR and accuracy.

Parameters	Calibration				Validation			
	AO	AO-SO	AO-CO	AO-RO	AO	AO-SO	AO-CO	AO-RO
True positive rate (TPR)	0.93	0.98	1.00	0.98	0.6	1.00	1.00	0.89
False negative rate (FNR)	0.07	0.02	0.00	0.02	0.4	0.00	0.00	0.11
Positive predictive value (PPV)	1.00	1.00	0.95	0.98	1.00	1.00	0.90	0.89
False discovery rate (FDR)	0.00	0.00	0.05	0.02	0.00	0.00	0.10	0.11
Accuracy	0.98				0.93			

**Table 3 foods-11-01134-t003:** Fatty acid composition and viscosity of pure AO, SO, CO and RO.

Samples	Avocado Oil				Soybean Oil	Corn Oil	Rapeseed Oil
Place of Origin	France	France	New Zealand	Mexico	China	China	China
Viscosity (mPa s)	61.56 ± 1.33	61.50 ± 0.70	61.22 ± 0.80	60.12 ± 0.61	48.20 ± 0.74	50.96 ± 0.24	56.56 ± 0.28
Fatty acids (%)							
Myristic acid (C14:0)	0.03 ± 0.00	0.04 ± 0.00	0.04 ± 0.00	0.04 ± 0.00	0.07 ± 0.00	0.04 ± 0.00	0.06 ± 0.00
Palmitic acid (C16:0)	11.84 ± 0.03	10.66 ± 0.01	11.47 ± 0.01	12.76 ± 0.01	10.66 ± 0.03	11.21 ± 0.01	4.67 ± 0.02
Palmitoleic acid (C16:1)	1.69 ± 0.00	1.82 ± 0.01	1.93 ± 0.00	2.53 ± 0.00	0.07 ± 0.00	0.09 ± 0.00	0.22 ± 0.00
Stearic acid (C18:0)	2.44 ± 0.02	2.31 ± 0.02	2.33 ± 0.01	2.25 ± 0.01	4.19 ± 0.01	2.06 ± 0.01	1.85 ± 0.03
Oleic acid (C18:1n9c)	69.2 ± 0.05	70.91 ± 0.01	71.03 ± 0.03	64.49 ± 0.04	24.09 ± 0.04	28.19 ± 0.02	58.14 ± 0.04
Linoleic acid (C18:2n6c)	13.24 ± 0.02	12.76 ± 0.02	11.77 ± 0.02	16.25 ± 0.02	53.39 ± 0.03	56.84 ± 0.02	20.24 ± 0.02
Arachidic acid (C20:0)	0.35 ± 0.00	0.32 ± 0.00	0.34 ± 0.00	0.32 ± 0.00	0.37 ± 0.00	0.34 ± 0.00	0.62 ± 0.00
γ-Linolenic acid (C18:3n6)	ND	ND	ND	ND	0.05 ± 0.00	0.04 ± 0.00	0.59 ± 0.00
cis-11-Eicosadienoic acid (C20:1)	0.27 ± 0.00	0.26 ± 0.00	0.27 ± 0.00	0.26 ± 0.00	0.20 ± 0.00	0.25 ± 0.00	ND
α-Linolenic acid (C18:3n3)	0.65 ± 0.00	0.52 ± 0.00	0.59 ± 0.00	0.93 ± 0.00	6.37 ± 0.01	0.58 ± 0.00	7.78 ± 0.03
cis-11,14-Eicosadienoic acid (C20:2)	ND	ND	ND	ND	ND	ND	0.12 ± 0.00
Behenic acid (C22:0)	0.22 ± 0.00	0.28 ± 0.00	0.18 ± 0.00	0.17 ± 0.00	0.40 ± 0.00	0.22 ± 0.00	0.35 ± 0.00
cis-11,14,17-Eicosatrienoic acid (C20:3n3)	ND	ND	ND	ND	ND	ND	5.09 ± 0.02
cis-5,8,11,14,17-Eicosapentaenoic acid (C20:5n3)	0.09 ± 0.00	0.12 ± 0.01	0.08 ± 0.00	ND	0.13 ± 0.00	0.14 ± 0.00	ND
Nervonic acid (C24:1n9)	ND	ND	ND	ND	ND	ND	0.28 ± 0.00
Saturated fatty acids (SFA)	14.87 ± 0.05	13.62 ± 0.03	14.36 ± 0.03	15.54 ± 0.02	15.69 ± 0.03	13.87 ± 0.02	7.55 ± 0.06
Monounsaturated fatty acids (MUFA)	71.16 ± 0.05	72.99 ± 0.02	73.22 ± 0.03	67.28 ± 0.04	24.36 ± 0.04	24.36 ± 0.02	58.65 ± 0.04
Polyunsaturated fatty acids (PUFA)	13.97 ± 0.02	13.39 ± 0.02	12.43 ± 0.02	17.18 ± 0.03	59.95 ± 0.04	57.60 ± 0.02	33.81 ± 0.06

Note: ND: not detected.

## Data Availability

Data is contained within the article or Appendix A.

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
