# Peer review of "Rapid Detection of Avocado Oil Adulteration Using Low-Field Nuclear Magnetic Resonance"

_foods, 2022, doi:10.3390/foods11081134_

Round 1

Reviewer 1 Report

The manuscript was well written, and it was also well organized

The scientific content of the presented manuscript is very relevant

However, authors must provide the data set used do build PCA, SIMCA and PLS models, it must be provided as a Microsoft Excel spreadsheet in supporting information.

Author Response

1.The manuscript was well written, and it was also well organized.

Response: Thank you for the comment.

2.The scientific content of the presented manuscript is very relevant.

Response: Thank you for the comment.

3.However, authors must provide the data set used to build PCA, SIMCA and PLS models, it must be provided as a Microsoft Excel spreadsheet in supporting information.

Response: Thanks for your suggestion. The data set used to build PCA, SIMCA and PLS models was provided as a Microsoft Excel spreadsheet in supporting information. The Microsoft Excel spreadsheet was named as “The data set used to build models” 

Reviewer 2 Report

The paper describes an application of TDNMR, a benchtop Nuclear Magnetic Resonance relaxometer that is suitable for quality control of food matrices, to the detection of adulteration in vegetable oils, in particular the avocado oil.

The scope of the work is relevant, and so is the interest of the readership toward quality control of foods and fraud detection, since huge economic loss and social issues are usually associated with this problem. Avocado oil is considered a delicacy, so it reaches a high price in the market. Suitable analytical techniques that would safeguard this production attract growing interest.

The experimental plan and number of replicates planned could have been sufficient, and the technique can be considered sufficiently responsive for detecting high percentage of alternative oils. However, the main drawbacks are, in my opinion, that samples have been pseudoreplicated (replicate analyses derive from the same batch of oil, if I understand well) and that the analysis of T2 distributions is not complete and many other features of NMR relaxation distribution profiles need to be addressed and explained. Moreover, Figures and figure captions are scarcely informative (and not self explanatory), and methods (especially NMR acquisition) was not detailed, and should be revised thoroughly. English should be revised, since there are several sentences that are difficult to read and grammatical errors.

In more detail:

  1. More literature should be cited and explained in Introduction to present how NMR relaxometry measurements of oils can be interpreted and related to lipid structures.
  2. Authors refer to as "peaks" when describing the result of NMR relaxometry distributions of relaxation times. However, that are not peaks but relaxation components.
  3. In Paragraph 2.2 the experimental acquisition parameters should be better detailed. Please define interpulse delays used in CPMG, sample temperature during acquisition, temperature equilibration time of the sample before acquisition in the NMR probe, measured T1 (NMR longitudinal relaxation time) of samples.
  4. Please describe briefly the methylation method [Amit et al, reference 23] cited at line 100, page 3.
  5. At line 153, page 4 it is stated " samples adulterated with higher level of SO and CO were observed to relax more slowly", which is not actually true. For example, in SO mixtures, T23 component relaxes more slowly, but T21 relaxes faster in mixtures and slower in pure SO: why? Similarly in CO mixtures T23 seems to relax more slowly as a function of increasing adulteration, but T22 has a somewhat random behavior and has a higher value in 40% than in 60% CO mixture. Authors must discuss differences in all relaxation components, and explain relating to structural features.
  6. It is not clear whether T2w is the monoexponential deconvolution of multiexponential signal decay or if this is due to a real monoexponential decay.
  7. Figure captions should be self-explanatory. For example, Figure S1 is very difficult to interpret and no indication is given to the reader in caption. What do color represent, and what unit is used in the color bar? What do letters A, B, C represent? What software has been used to generate it? What is T2w?
  8. Please provide all data (as supplementary materials) concerning P21, P22 and P23. What do authors mean when they say "generally decreased"? What happens when they do not decrease?
  9. PCA scores plot in Figure 2 report a very borderline explanation of variance (50.3% of variance explained). The model is not working very good. I would be careful in drawing conclusion based on these data
  10. Supplementary Materials should report at least the title of the paper in the first page.

Reviewer 3 Report

This manuscript proposes the application of low-field NMR to study avocado oil adulterations with cheaper vegetable oils such as soybean or corn oils. The paper may be of interest dor the Foods readers but various drawbacks have to be addressed thoroughly.

General comments

My first surprise was When I examined the paper, my first surprise was that the authors have not used the NMR spectra as the sample fingerprints to proceed to the characterization and classification studies. Instead, they have used relaxation data for the compounds. I do not know if this is a very widespread option. I have seen in the literature various papers using such a type of data; anyway, as far as I know, the analysis of NMR spectra seems to be more common. Perhaps the authors should discuss this topic. Wouldn't it be possible, in this case, to also use NMR spectra?

From the point of view of data quality, when checking the plot of scores of Fig 2, I guess that PC1 mainly explains a kind of data drift. For instance, AO samples are scattered from left to right, Why? Something to do with the measurement sequence/order? If so, samples should have been measured randomly. This issue might seriously compromise the performance of the descriptions.

Subplots c and e in Figure 5 suggest that compositional profiles of fatty acids are much better descriptors of oils than LF-NMR: Samples belonging to each oil type are grouped in a much more compact way without overlapping among classes. If so, the election of LF-NMR is doubtful. Authors have to convince the audience that the proposed approach is scientifically sensible and, in practice, that it is more reliable and efficient than the existing methods.

Related to the previous point, the pros and cons of this approach concerning other existing methods should be discussed extensively.

A table with T2W, T21s, … data of the pure AO (the 4 brands), SO, CO, and RO should be given in the supplementary material.

Introduction: I have checked the papers published on this topic and some related recent applications are missing (see Alonso, Tang, Turrini, Martin-Torres, Wang, Sotelo, and others). Besides, I think that the description is quite narrow-minded on avocado oil adulterations and perhaps a broader approximation facing other authentication cases (olive oil case has been extensively treated).

Conclusions are poor. This section must be improved substantially.

Specific comments

Abstract: The concept “relative contribution” should be explained. SVR should be defined.

Plots in Fig 3 show the performance of the assignation to the different classes. However, I suggest including an illustrative example with a complementary plot of Q residual vs Hotelling’s t2 to see the sample distribution in 2D.

How the values of the DCcrit parameter are established? It seems to be quite arbitrary.

Table 1: SMICA should be SIMCA? The meaning of Member, Factor, R2X, Q2, and AUC should be explained in the table footer. It is not clear how many PCs are used for each SIMCA model.

The authors have to revise the paper thoroughly to correct some grammatical errors. Just some examples: Line 30: phenolic needs a substantive: Species, compounds, or analytes. Line 36: to confirm should be confirming. Line 46: Whereas. Line 49: were applied. Line 54: in addition, the. 

Round 2

Reviewer 3 Report

The paper has been improved substantially. In my opinion is OK.